# Effect of Deep Dormancy Temperature Cultivation on Meat Quality of Crucian Carp (*Carassius auratus*)

**DOI:** 10.3390/foods12040792

**Published:** 2023-02-13

**Authors:** Yin Zhang, Linguo Wang, Yunlong Mu, Qing Zeng, Jianlin Jia, Pengcheng Zhang, Zhongli Pan

**Affiliations:** 1Meat Processing Key Laboratory of Sichuan Province, Chengdu University, Chengdu 610106, China; 2Department of Biological and Agricultural Engineering, University of California, One Shields Avenue, Davis, CA 95616, USA

**Keywords:** critical dormancy, crucian carp, blood glucose, texture, nucleotides

## Abstract

To extend the survival of crucian carp (*Carassius auratus*) during transportation, the effect of deep dormancy temperature (DDT) cultivation on the crucian carp was investigated by measuring the respiratory rate, survival time, and effect of cooling speed on the meat quality. The results of the respiratory rate and survival time indicated that the DDT of the crucian carp was 1.6 °C. The cooling speed had a significant (*p* < 0.05) influence on the quality of the crucian carp meat, with a faster cooling speed resulting in a lower pH, L* value, a* value, gumminess, springiness, cohesiveness, stickiness, chewiness, CMP, and UMP content for the crucian carp meat, thus resulting in a lower sensory score for the crucian carp meat. A possible reason for the decrease in the quality of the crucian carp meat is that the faster cooling speed led to a strong stress response and higher anaerobic metabolism in the crucian carp. This can be supported by the contents of the blood glucose and lactic acid in the crucian carp treated with higher cooling speed being significantly (*p* < 0.05) higher than that of the control. Combining all the results of the cooling speed on the eating quality of the crucian carp meat, a cooling speed of 2 °C/h followed by 1 °C/h was suggested for the survival of crucian carp in transit.

## 1. Introduction

Fresh aquatic products are preferred by consumers due to their good taste and high nutritional value [1,2]. However, regional differences in fish cultivation mean that live fish have to be transported across regions to meet the demand of markets in different areas [3]. According to statistics from the Ministry of Agriculture and Rural Affairs of the People’s Republic of China, the transregional transportation volume of live freshwater fish was 25,973,200 tons in China in 2021 [4]. During the transportation of live freshwater fish, the water temperature, oxygen supply, and microbial contamination will lead to a high mortality rate [5]. It has been estimated that about 10% of live fish die due to improper transportation methods in China [3]. Such a large mortality rate not only causes a serious loss to agriculture and fisheries, but also raises the cost of freshwater fish. In addition, due to the stress response of live fish to an improper transportation environment, the quality of the meat greatly decreases [6,7]. In order to decrease the mortality rate and maintain the quality of live fish during transportation, researchers have explored various methods to improve transportation efficiency. 

Wang et al. [8] studied the effects of water temperature (25 °C, 4 °C) and transportation methods (with or without water) on the survival time of the Amur Sturgeon (*Acipenser schrencki*), and found that it stayed alive for 32 h at 4 °C without water, and experienced no significant change in meat quality of 32 h. Spotted sea bass (*Lateolabrax maculatus*) was transported at 4 °C with water, with less water, and without water for 9 h, and the survival rate was 49.11%, 15.60%, and 23.96%, respectively, after transportation. It was concluded that the method of waterless transportation was feasible for spotted bass, but the transportation time should be controlled within 3 h [9]. Nie et al. [10] investigated the effects of low temperature and waterless transportation on the biochemistry and morphology of turbot (*Psetta maxima*) and found that transportation at 2 °C slowed down the metabolic rate of the turbot and maintained it in a dormant state. Low-temperature transportation was also effective for Giant River Prawns (*Macrobrachium rosenbergii*), whose survival rate could remain 100% when it was kept at 20 °C for 24 h with a mass ratio of shrimp to water of 1:10 [11]. Sea bass (*Lateolabrax maculatus*) muscle showed no significant changes after being kept alive and transported at 12 °C for 72 h, and the eating quality of its muscle was basically the same as that of the control [9]. These studies indicated that low-temperature transportation can prolong the survival time of fish or shrimp without a serious influence on their eating quality, but few investigations have been performed to evaluate the effect of deep dormancy temperature on the meat quality of crucian carp (*Carassius auratus*).

Crucian carp (*Carassius auratus*) is one of the most widely consumed freshwater fishes in China. It not only has tender meat quality, but also some beneficial medical effects [12]. In 2020, the output of crucian carp in China reached 2.75 million tons, with the aquaculture areas mainly distributed in Jiangsu and Hubei [13]. Due to the continuous growth of the domestic economic level in recent years, the market demand for crucian carp has continued to grow in China, but the production and sales of crucian carp have a large geographical span. This situation necessitates the live transportation of crucian carp. A dormancy state of fish is considered to be when the breathing rate is lower than 20 times/min at a lower temperature [14]. The temperature at which the fish is alive but has the lowest breath rate is considered the deep dormancy temperature (DDT). Theoretically, a fish transported in deep dormancy will have a greatly decreased metabolic rate and activity, which will result in a lower mortality rate because of less bodily harm and energy metabolism. Recently, it was found that pork stored at a deep chill temperature (ice temperature) had superior meat quality to that stored at 4 °C [15,16]. It is necessary to evaluate the effect of DDT on the survival time of crucian carp and its meat quality, which may be helpful to improve the transportation efficiency of fish by extending its survival time and changing its eating quality less. Therefore, the aims of this investigation are to determine the DDT of the crucian carp and evaluate its effect on meat quality during cultivation. 

## 2. Materials and Methods

### 2.1. Materials

Live crucian carp (*Carassius auratus*) were purchased from Tongwei Sanlian Aquatic Products Trading Center (Chengdu, China), with weights of 230 ± 10 g and body lengths of 20 ± 2 cm. For all the experiments, intact, vigorous, and healthy crucian carp were used and cultured in polyethylene tank (0.5 × 0.4 × 0.3 m), with running water (20 ± 1 °C, pH 7.0 ± 0.5, the average dissolved oxygen was 6.0 mg/L), for 48 h before the experiments. According to the decision of the Animal Ethics Committee of Chengdu University, the fish used in experiments were washed with running water, dried with filter paper, and killed by knocking the head. Collect blood samples from the fish tail vein and the fish back muscle for biochemical analysis. Guanosine 5′-monophosphate disodium salt hydrate (5′-GMP-Na2, with purity 99.0%), cytidine 5′-monophosphate (5′-CMP, with purity 99.0%), uridine 5′-monophosphate disodium salt (5′-UMP-Na2, with purity 99.0%), 5′-inosinic acid disodium salt hydrate (5′-IMP-Na2, with purity 99.0%), and adenosine 5′-monophosphate disodium salt (5-AMP-Na2, with purity 99.0%) were purchased from Sigma-Aldrich (St. Louis, MO, USA); potassium dihydrogen phosphate, trichloroacetic acid, hydrochloric acid, and other chemical reagents were purchased from Chengdu Cologne Chemical Co., Ltd. (Chengdu, China); a lactate kit and glucose kit were purchased from Nanjing Jiancheng Bioengineering Institute (Nanjing, China).

### 2.2. Determination of DDT of Crucian Carp

Twelve crucian carp (purchased on 24 November 2020, from Tongwei Sanlian Aquatic Products Trading Center with weights of 230 ± 10 g and body lengths of 20 ± 2 cm) were used to perform six experiments (two fish per batch). For each batch, the cultured crucian carp were placed in a low-temperature thermostat (DC-4015, Shanghai Pingxuan Scientific Instrument Co., Ltd., Shanghai, China) and cooled at a speed of 3 °C/h from 20 °C to 5 °C, and then cooled from 5 °C to 0 °C at a speed of 1 °C/h. The appearance, respiratory rate, presence or absence of bleeding, and active state of the crucian carp were observed every 60 min and recorded at each cultivating temperature (0, 0.5, 1, 2, 3, 4, 5, 8, 11, 14, 17, or 20 °C). According to Zeng, Shen, and Chen [14], a breathing rate lower than 20 times/min is considered a dormant state. The results of this experiment can reveal the temperature range of the crucian carp in a dormant state. 

In order to obtain DDT, 12 crucian carp (purchased on 16 December 2020, from Tongwei Sanlian Aquatic Products Trading Center, with weight 230 ± 10 g and body length 20 ± 2 cm) were cooled again and six experiments (two fish per batch) were performed. For each batch, the same cooling speed was used—a speed of 3 °C/h from 20 °C to 5 °C, a speed of 1 °C/h from 5 °C to 2 °C, and then a speed of 0.2 °C/h from 2 °C to 0 °C. The breathing rate of the crucian carp was observed for 60 min and recorded every 10 min at each cultivating temperature (0, 1, 1.6, 2, 3, 4, 5, 8, 11, 14, 17, or 20 °C). 

### 2.3. Determination of Survival Time

To investigate the effect of cultivation temperature on the survival time of the crucian carp, 66 crucian carp were purchased on 8 January 2021, from Tongwei Sanlian Aquatic Products Trading Center, with weights of 230 ± 10 g and body lengths 20 ± 2 cm. Six crucian carp (two fish per batch) were cultivated at 1.0, 1.6, 2.0, 3.0, 4.0, 5.0, 8.0, 11.0, 14.0, 17.0, and 20.0 °C, respectively. The temperature was maintained by a thermostat (DC-4015, Shanghai Pingxuan Scientific Instrument Co., Ltd.). As the respiratory rate of the crucian carp was less than 5 breaths/min, the fish were taken out from the thermostat, washed with running water, dried with filter paper, and killed by knocking the head. The average survival time of the crucian carp was recorded as the survival time for each group.

### 2.4. Cooling Procedure 

The cooling process affects the physiology and eating quality of fish [17]. In order to choose a proper cooling procedure for the cultured crucian carp to suit the culturing temperature decreasing from 20 to DDT, according to Zeng, Chen, and Shen [17], four experiments (S1, S2, S3, and S4) were performed. In Experiment S1, the cultured fish was cooled from 20 °C to 5 °C at a speed of 5 °C/h, and then cooled from 5 °C to DDT at a speed of 1 °C/h. In Experiment S2, the cultured fish was cooled from 20 °C to 5 °C at a speed of 3 °C/h, and then cooled from 5 °C to DDT at a speed of 1 °C/h. In Experiment S3, the cultured fish was cooled from 20 °C to 5 °C at a speed of 2 °C/h, and then cooled from 5 °C to DDT at a speed of 1 °C/h. In Experiment S4, crucian carp cultured at 20 ± 1 °C without cooling processing was used as the control. One hundred and forty-four crucian carp were purchased on 5 February 2021, from Tongwei Sanlian Aquatic Products Trading Center, with weights of 230 ± 10 g and body lengths of 20 ± 2 cm. For each experiment, 12 crucian carp were used to perform six experiments (two fish per batch), and three repetitions (12 crucian carp per repetition) were evaluated for each experiment.

### 2.5. Determination of Blood Glucose of Crucian Carp

To evaluate the effect of cultivation at DDT on stress in crucian carp [18], the blood glucose concentration of the crucian carp was determined with a blood glucose kit (Nanjing Jiancheng Bioengineering Institute and a UV-5200 ultraviolet spectrophotometer (Shanghai Fiber Inspection Instrument Co., Ltd., Shanghai, China). As the crucian carp were cooled to DDT and maintained at DDT for 1 h. The crucian carp were taken out from the thermostat, washed with running water, and dried with filter paper. Before being killed by knocking the head, the blood was collected from the tail vein of each crucian carp, and the processed serum was stored at −20 °C before determination. Twenty microliters of the serum were loaded into a blood glucose kit (Nanjing Jiancheng Bioengineering Institute) to determine the blood glucose content. Triplicate determinations were performed for each sample and the content of blood glucose was expressed as mmol/L.

### 2.6. Determination of Blood Lactic Acid of Crucian Carp

To analyze the effect of cultivation at DDT on stress in crucian carp [18], the lactic acid concentration of crucian carp was determined with a lactic acid kit (Nanjing Jiancheng Bioengineering Institute), and a UV-5200 ultraviolet spectrophotometer (Shanghai Fiber Inspection Instrument Co., Ltd.). As the crucian carp were cooled to DDT and maintained at DDT for 1 h. The crucian carp were taken out from the thermostat, washed with running water, and dried with filter paper. Before being killed by knocking the head, the blood was collected from the tail vein of each crucian carp, and the processed serum was stored at −20 °C before determination. Twenty microliters of the serum were loaded into the lactic acid kit (Nanjing Jiancheng Bioengineering Institute) to determine the lactic acid concentration. Triplicate determinations were performed for each sample, and the lactic acid content was expressed as mmol/L.

### 2.7. Determination of pH 

The pH of the crucian carp meat was determined according to Annamalai et al. [19] and Hu et al. [20] with slight modification. Ten grams of dorsal muscle blocks of the crucian carp meat were minced and homogenized in 100 mL of ultra-pure water (Arum-mini, Jinan Ouweiteng Biotechnology Co., Ltd., Jinan, China) for 30 min. Then, the pH of the filtrate was determined with a pH meter (PHS-3G, Shenzhen 3NH Technology Co., Ltd., Shenzhen, China). Triplicate determinations were performed for each sample.

### 2.8. Determination of Water-Holding Capacity (WHC)

The WHC of the crucian carp meat was determined according to Ocano-Higuera et al. [21] and Zhang et al. [22] with slight modification. Ten grams of dorsal muscle blocks of the crucian carp meat were wrapped with a double filter paper (Civil Filter Paper Factory, Fushun, China) and placed in a 50-mL centrifuge tube and centrifuged (TLG-1650, Sichuan Shuke Instrument Co., Ltd., Chengdu, China) at 1000 r/min for 15 min at room temperature (20 ± 1 °C). Triplicate determinations were performed for each sample. The *WHC* was calculated as follows:(1)$$\mathrm{WHC}\;(\%)\;=\;\frac{M-m}{M}\;\times \;100$$
where *M* is the mass of fish meat before centrifugation; *m* is the mass of fish after centrifugation.

### 2.9. Determination of Cooking Loss

The cooking loss of the crucian carp meat was analyzed according to Rodríguez et al. [23] with slight modification. Briefly, 10 g of dorsal muscle blocks of the crucian carp meat was placed in a self-sealing bag (Zhilong Packaging Company, Taizhou, China) in a 75 °C water bath (HH-6 thermostatic water bath, Changzhou Aohua Instrument Co., Ltd., Changzhou, China) for 15 min until the core temperature reached 60 °C and kept at 60 °C for 30 min, after which it was stood for 15 min to cool naturally to room temperature (26~27 °C). The surface juices were blotted with filter paper. Triplicate determinations were performed for each sample. *Cooking loss* was calculated as follows:(2)$$\mathrm{Cooking}\;\mathrm{loss}\;(\%)\;=\;\frac{H\;-\;h}{H}\;\times \;100$$
where *H* is the mass of fish meat before cooking; *h* is the mass of fish after cooking.

### 2.10. Determination of Color Difference

The color of the dorsal muscle blocks of the crucian carp meat was determined according to Feng et al. [24] and Zhang et al. [25] with slight modification. An NH310 colorimeter (Shenzhen 3NH Technology Co., Ltd.) was used to measure the L*, a*, and b* values of the crucian carp meat. Six determinations were performed for each sample. 

### 2.11. Determination of Muscle Fiber Diameter

To evaluate the tenderness of the crucian carp meat, the muscle fiber diameter of the crucian carp was determined according to Hou [26], with a slight modification. Briefly, the crucian carp dorsal muscle blocks (2.0 cm × 2.0 cm × 1.0 cm) were fixed in 10% neutral formalin solution and stained with the hematoxylin–eosin staining method. The longest muscle fiber cross-section was measured with a CX33 microscope (Shandong Shiyi Optoelectronics Technology Co., Ltd., Shanghai, China) at a 10 × 40 magnification. Fifteen muscle fibers from each sample were measured, and the average value was calculated from the vertical distance of its midpoint. 

### 2.12. Determination of Texture Profiles of Crucian Carp

To compare the texture profiles (hardness, adhesiveness, springiness, cohesiveness, gumminess, chewiness, and resilience) of the crucian carp meat, a texture profile analysis (TPA) was performed according to Zhang et al. [27] with slight modification. Nine pieces of dorsal muscle blocks of the crucian carp meat (2.0 cm × 2.0 cm × 1.0 cm) were sampled and the texture profile determined with a TA.XT. Plus texture analyzer (Stable Micro System Inc., Surrey, UK). The instrument parameters were set as follows: a P36/R probe was used, the probe drop distance was 25 mm and the speed was 2 mm/s before the test, the speed was 1 mm/s during the test, the speed was 5 mm/s after the test, the strain was 40%, the trigger force was 5 g, the interval between the two compression times was 5 s, and the probe return speed was 5 mm/s after the test. 

### 2.13. Determination of Nucleotides

To analyze the effect of cultivating at DDT on the nucleotides in the crucian carp meat, the content of nucleotides in the crucian carp meat was analyzed according to Zhang, Zhang, Li, Guo, Jia, Zhang, Wang, Xia, Qian, and Peng [27] with some modification. Briefly, 10 g of dorsal muscle blocks of crucian carp meat was pulverized with an ultra micro pulverizer (IKA company, Staufen, Germany) and homogenized in a 50-mL plastic centrifuge tube with 40 mL of cold 5% TCA and placed in a BCD- 649WDCE refrigerator (Haier Company, Qingdao, China) at 4 °C for 1 h. The homogenate was centrifuged with a TLG-1650 centrifuge (Sichuan Shuke Instrument Co., Ltd.) at 5000 r/min for 10 min, and the supernatant was collected and transferred to a 100-mL beaker. The precipitate was reextracted and the supernatants combined. The pH of the supernatant was adjusted to 6.5 with 3 mol/L KOH solution, and then filtered with qualitative filter paper (Civil Filter Paper Factory) into a 50-mL volumetric flask. The filtrate was diluted with ultra-pure water to 50 mL. The solution was filtered again with a 0.22 μm filter membrane (Xinya Purification Device Factory, Shanghai, China), and the filtrate was analyzed with an Agilent 1100 high-performance liquid chromatograph (Agilent company, Palo Alto, CA, USA), which was equipped with Hypersil ODS2-C18 (4.6 mm × 250 mm, 5.0 μm) and a UV detector. Quantitative determination was performed by the external standard method, and triplicate experiments were done for each sample. Nucleotide contents were expressed as mg of 5′- mononucleotide per 100 g of muscle. 

### 2.14. Sensory Evaluation

To assess the effect of cultivation at DDT on the eating quality of crucian carp meat, a sensory evaluation was performed according to Que [28] with some modifications. The crucian carp dorsal muscle meat pieces (2 cm × 2 cm × 1 cm) were put into a cooking bag (Zhilong Packaging Company) and heated in a HH-6 thermostatic water bath (Changzhou Aohua Instrument Co., Ltd.) at 96 °C for 5 min. When the carp meat pieces were warm (55 ± 5 °C), seven pieces of fish meat for each sample were served for panelists to score. Nine sensory panelists were selected and trained [29,30] to use quantitative descriptive analysis [31] to assess the appearance, odor, taste, and sensory texture of the fish meat. The generation and selection of scoring criteria were carried out by open discussion in five sessions prior to the experiments. The scoring criteria retained for the sensory evaluation are described in Table 1. A non-structured scoring scale was used, where 0 and 10 were the lowest and the highest, respectively. The sensory evaluation was carried out in five sessions for each group of fish meat using a complete block design, where each panelist evaluated one kind of fish meat from each test three times in each session. Triplicate sensory evaluations were performed for each group.

### 2.15. Data Analysis

The results were analyzed by ANOVA at a significance level of 5% (H0: *p* < 0.05). The comparison of means was analyzed by Fisher’s LSD tests using the SAS 9.1 statistical package (SAS Institute, Inc., Raleigh, NC, USA).

## 3. Results and Discussion

### 3.1. DDT and Temporary Rearing Conditions of Crucian Carp

#### 3.1.1. DDT of Crucian Carp

The DDT of the crucian carp was obtained based on the range of dormancy temperature of the crucian carp. The data in Table 2 indicate that the respiratory rate of the crucian carp decreased with a reduction in the cultivating temperature. The respiratory rate of the crucian carp started to be irregular when the water temperature dropped to 5.0 °C. As the cultivation temperature reached 2.0 °C, the crucian carp did not swim and stayed still at the bottom of the tank. When the cultivation temperature was between 2.0 °C and 1.0 °C, the crucian carp exhibited weak breathing and could respond to stimulation. When the cultivation temperature was 0.5 °C, the crucian carp had weak breath and made no response to stimulation. When the crucian carp were cultivated at 0 °C, they were bending into an arched shape, had very weak breathing, and died within a short time. According to the behavior of the crucian carp and the respiratory rate below 20 times/min [14], the critical dormancy temperature of the crucian carp should be 2.0 °C, and the range of the dormancy temperature of the crucian carp should be between 1.0 °C and 2.0 °C. 

To further obtain the DDT, the respiratory rate of the crucian carp was determined repeatedly. The results (Figure 1) show that the respiratory rate of the crucian carp decreased with the reduction in cultivation temperature, which showed a similar tendency as that in Table 1. The respiratory rate of the crucian carp cultivated at different temperatures (0.0, 1.0, 1.6, 2.0, 3.0, 4.0, 5.0, 8.0, 11.0, 14.0, 17.0, or 20.0 °C) for different times (0, 10, 20, 30, 40, 50, or 60 min) indicated that the respiratory rate showed no significant difference (*p* > 0.05) when the crucian carp was cultivated at each temperature for more than 40 min. The respiratory rate of the crucian carp was the lowest and most stable when it was cultivated at 1.6 °C. This result is consistent with that of Wang et al. [32], who reported that the range of the dormancy temperature of freshwater California perch was between 0 and 2.0 °C.

#### 3.1.2. The Survival Time of Crucian Carp at Different Temperatures

To compare the effect of cultivation temperature on the survival time of crucian carp, the survival time at 1.0, 1.6, 2.0, 3.0, 4.0, 5.0, 8.0, 11.0, 14.0, 17.0, or 20.0 °C was determined (Figure 2). The results in Figure 2 indicate that the survival time of the crucian carp increased significantly (*p* < 0.05) as the cultivation temperature increased from 1.0 °C to 1.6 °C, and then decreased significantly (*p* < 0.05) as the cultivation temperature increased from 1.6 °C to 20.0 °C. These results indicate that the crucian carp can remain alive for 251.3 h at DDT (1.6 °C), which is an increase of 95.3% compared with crucian carp cultivated at 20 °C (control), and 11.7% longer than crucian carp cultivated at 2.0 °C (critical dormancy temperature). Therefore, cultivation at DDT is more beneficial to the survival of the crucian carp than that at the critical dormancy temperature (2.0 °C) or the control temperature (20 °C).

The fish cultivated at DDT showed the lowest respiratory rate (Figure 1). This suggests that the crucian carp exhibited less physiological activity and a lower metabolic level [33]. The lower physiological activity of the crucian carp can decrease the rate of injury caused by collisions with each other or with the vessel wall during cultivation. A higher injury rate will increase the death rate of the crucian carp due to skin infection. The lower metabolic level will make the crucian carp resist stress reactions to the environment and maintain bodily functions, and thus survive for a long time [6]. These might be the main reasons why crucian carp cultivated at DDT show a longer survival time than those cultivated at other temperatures. A similar reason was inferred when the yellow catfish (Banded Catfish; Pelteobagrus fulvidraco) was cultivated at a low temperature (2.0 °C) in an anhydrous dormancy state and pure oxygen environment for 24 h [34].

#### 3.1.3. Effects of Different Cooling Rates on Survival Time of Crucian Carp

The reduction rate of water temperature is very important for the survival rate and meat quality of fish [35]. Too fast a cooling speed will trigger a strong stress reaction in fish, which causes physical injury and leads to a decline in meat quality [35,36]. Therefore, for practical applications, it is necessary to establish an optimal cooling process for the cultivation of crucian carp at DDT. To achieve this goal, the effects of different cooling speeds on survival time and meat quality of the crucian carp were compared. 

The data in Figure 3A indicate that the survival time of the crucian carp in group S3 (272 ± 7 h) was significantly (*p* < 0.05) higher than that in group S1 (199 ± 6 h) and S2 (246 ± 9 h), and the shortest survival time was observed in the S1 group (199 ± 6 h). The main difference in the cooling process of groups S1, S2, and S3 was in the cooling stage from 20 °C to 5 °C: the cooling speed of groups S1, S2, and S3 was 5 °C/h, 3 °C/h, and 2 °C/h, respectively. These results indicate that a lower cooling speed resulted in a longer survival time. This is consistent with Qin [36], who found that a lower cooling speed could prolong the survival time of the Pengze Crucian Carp. 

The cooling speed of group S3 was lower compared to groups S1 and S2. The cooling speed influences the stress response of fish [36]: a higher cooling speed makes it hard for fish to adapt to rapid changes in temperature and accelerates energy consumption, thus resulting in a significant reduction in survival time [36]. Therefore, the difference in cooling speed may be the main reason for the difference in survival time of groups S1, S2, and S3. In order to confirm the effect of cooling speed on the stress reaction of the crucian carp, the effects of cultivation at DDT on the edible quality and physiology and biochemistry of crucian carp were analyzed.

### 3.2. Effects of Cultivation at DDT on the Edible Quality of Crucian Carp

#### 3.2.1. The pH, WHC, and Cooking Loss of Crucian Carp Meat

To evaluate the effect of cooling speed on the meat quality of the crucian carp, the effects of different cooling speeds on WHC (Figure 3B), pH (Figure 3C), and cooking loss (Figure 3D) of the crucian carp meat were determined. The results in Figure 3B–D indicate that WHC and the cooking loss of the crucian carp meat were not significantly (*p* > 0.05) influenced by the cooling speed; the pH of group S4 was significantly (*p* < 0.05) higher than that of groups S1 and S2, but there was no significant (*p* > 0.05) difference between groups S4 and S3. The WHC, pH, and cooking loss of group S3 were similar to those of group S4 (control). These results suggest that the cooling speed of group S3 (a speed of 2 °C/h, followed by 1 °C/h) had little influence on the meat quality of the crucian carp. 

WHC is the ability of a muscle to retain water when subjected to external forces, and is affected by the deformation and degradation of the protein structure of myogenic fibers [37]. Cooking loss refers to the loss of water and soluble substances in muscle after steaming [38]. The pH of meat is used for characterizing the acid content. The WHC and cooking loss of the crucian carp meat changed little, but the pH of the crucian carp meat decreased significantly (*p* < 0.05) for groups S1 and S2. The increase in anaerobic metabolism in the process of keeping alive may be the main reason for the decrease in the pH of muscle [39], but the anaerobic metabolism had no significant (*p* > 0.05) influence on the denaturation of meat protein. This may be why the WHC and cooking loss of the crucian carp meat in groups S1, S2, S3, and S4 (control) were not significantly (*p* > 0.05) different.

#### 3.2.2. The Color of Crucian Carp Meat

Meat color is an important indicator for the consumer to judge the quality of fish [40]. The color indexes of the crucian carp meat in Figure 4 indicate that the L* values of groups S1 and S2 were significantly (*p* < 0.05) lower than those of groups S3 and S4, but there was no significant (*p* > 0.05) difference between groups S3 and S4. A similar significance can be discerned for a* value of the crucian carp meat. The b* values of groups S1, S2, and S3 showed no significant (*p* > 0.05) differences, and the b* values of groups S3 and S4 were similar (*p* > 0.05). These results suggest that a lower cooling speed from 20 °C to 5 °C is conducive to maintaining the meat color. The 2 °C/h followed by 1 °C/h cooling processing was superior to others to retain the meat color.

L* indicates the lightness of meat; a positive a* value represents red, while a positive b* value represents yellow [41,42]. The anaerobic metabolism increased with the fish stress reaction in the process of keeping it alive [39]. The cooling speeds of groups S1 and S2 were higher than that of group S3. High-speed cooling processing may induce a vehement stress reaction in the fish body, and increased anaerobic metabolism, which results in the accumulation of dark red blood in fish, and thus decreases the lightness and increases the b* value. This may be the reason for the differences in the L*, a*, and b* values of the crucian carp meat. The effects of cooling speed on the color of the crucian carp meat further supported the fact that the pH values of groups S1 and S2 were lower than those of groups S3 and S4. 

#### 3.2.3. Texture Profile of Crucian Carp Meat

Texture profiles are important indexes showing meat’s eating and processing quality [43]. The data in Table 3 indicate that there were no significant (*p* > 0.05) differences in the hardness of all groups; the gumminess, springiness, cohesiveness, stickiness, chewiness, and resilience of groups S3 and S4 were not significantly (*p* > 0.05) different. The springiness, cohesiveness, stickiness, and resilience of group S1 were similar to that of groups S2 and S3. There was no significant (*p* > 0.05) difference in the gumminess and chewiness of groups S1 and S2. These results indicate that the cooling speed influenced the texture profiles of the crucian carp meat: the crucian carp cooled at 2 °C/h followed by 1 °C/h (group S3) showed similar texture profiles to group S4 (control), which is consistent with the pH, L* value, and a* value of the crucian carp meat treated with different cooling speeds.

Texture profiles are indexes reflecting the mechanical properties of meat. The cooling speed of group S1 was faster than that of groups S2 and S3, which may induce a more vehement stress reaction in the crucian carp. The vehement stress reaction accelerated the more anaerobic metabolism of the crucian carp and promoted the consumption of glycogen and other materials in the crucian carp’s body. This may induce the texture profiles of the crucian carp muscle to change with the cooling speed. This speculation can be supported by the higher cooling speed (S1) resulting in the higher pH of the crucian carp meat, which may be induced by the lactic acid engendered by anaerobic metabolism during the cooling process (Figure 3C). 

#### 3.2.4. Umami-Enhancing Nucleotide Content of Crucian Carp Meat

Umami taste is conducive to nutritional intake among the elderly and patients decreasing their intake of sodium chloride and fat [44,45], and, as a result, is considered an important index to evaluate the eating quality of meat [15,46]. Umami nucleotides (CMP, UMP, GMP, IMP, and AMP) are very important to meat taste [15,42], and all of them can enhance the umami taste of the meat by combining monosodium glutamate with other umami ingredients [44,45]. In order to analyze the effect of cooling speed on the umami nucleotide content of the crucian carp meat, the contents of CMP, UMP, GMP, IMP, and AMP were determined (Table 4). The results in Table 4 indicate that there was no significant (*p* > 0.05) difference in the contents of GMP and IMP in all the groups; the contents of CMP and UMP of groups S2 and S3 were not significantly (*p* > 0.05) different from those of group S4. The contents of CMP and UMP of group S1 were significantly (*p* < 0.05) lower than those of group S4, but there was no significant (*p* > 0.05) difference among groups S1, S2, and S3. The content of AMP in group S1 was significantly (*p* < 0.05) higher than that in group S4, while there was no significant (*p* > 0.05) difference in the content of AMP among groups S2, S3, and S4. The difference in the content of the umami-enhancing nucleotides further supported the conclusion that the cooling speed influenced the eating quality of the crucian carp. 

In the cooling process, the crucian carp experiences a stress response and decomposes ATP to release energy to maintain body temperature. In this process, ATP was transformed into AMP through the catalysis of nucleoside-diphosphate kinase and nucleoside-triphosphate-adenylate kinase [47,48]. A higher cooling speed may result in more ATP being transformed into AMP. This may be the reason for the content of AMP of the groups following the order S1 > S2 > S3 > S4. UMP is transformed into CMP by consuming ATP and glutamine deamination. This may be a possible reason for the content of UMP and CMP in groups S1, S2, S3, and S4 showing similar significant differences. AMP can be decomposed into IMP under the catalysis of AMP deaminase. IMP was further decomposed into GMP under the catalysis of IMP dehydrogenase and GMP synthase [47,48]. The cooling speed of groups was in the order S1 > S2 > S3. The higher cooling speed may induce more ATP to be transformed into AMP and thus can result in more AMP being transformed into IMP and GMP. This may be why the contents of GMP and IMP in groups S1, S2, and S3 were higher than in group S4. 

#### 3.2.5. Sensory Evaluation of Crucian Carp Meat

To further evaluate the effect of the cooling speed on the eating quality of the crucian carp, a sensory evaluation was performed (Table 5). The data in Table 5 indicate that there was no significant (*p* > 0.05) difference in the appearance of groups S2, S3, and S4; the appearance of group S1 was significantly (*p* < 0.05) less fresh. The odor of group S4 was significantly (*p* < 0.05) stronger than that of groups S1 and S2; there was no significant (*p* > 0.05) difference in the odor of groups S3 and S4. The taste and sensory texture of groups S1 and S2, or S3 and S4 were not significantly (*p* > 0.05) different, but the taste and sensory texture of groups S1 and S2 were significantly (*p* < 0.05) lower than those of groups S3 and S4. The sensory texture of the groups was similar to the texture profiles of springiness and resilience. 

The L* and a* values of group S4 were significantly (*p* < 0.05) higher than those of groups S1, S2, and S3 (Figure 4A,B), which may be why the appearance of group S4 was significantly (*p* < 0.05) changed compared to the other groups. Group S4 did not experience anaerobic metabolism compared to the other groups, which may be the reason that the odor of group S4 was stronger than the other groups. The UMP and CMP contents of group S4 were significantly (*p* < 0.05) higher than group S1, but not significantly (*p* > 0.05) higher than group S2 and S3; the AMP content of group S4 was significantly (*p* < 0.05) lower than group S1, and the AMP contents of group S2 and S3 were similar (Table 4). The combined umami-enhancing effect of UMP and CMP was superior to that of AMP [49]. As umami is one of the key components of the savory taste of meat [50], this may be why the taste of groups S3 and S4 were significantly (*p* < 0.05) higher than the others. The results for sensory texture were similar to those for springiness and resilience, which further supported the results of the texture profiles. 

### 3.3. Effects of Cultivation at DDT on the Physiology and Biochemistry of Crucian Carp

#### 3.3.1. Blood Glucose Content of Crucian Carp

To confirm the effects of cooling speed on the stress response of the crucian carp, the blood glucose of the crucian carp was determined. The results of the blood glucose content in the crucian carp (Figure 5A) indicated that the content of blood glucose in the groups followed the order S1 > S2 > S3 > S4, with the content of blood glucose in group S1 being significantly (*p* < 0.05) higher than in the other groups, that group S4 was significantly (*p* < 0.05) lower than in the others, that groups S2 and S3 were significantly (*p* < 0.05) lower than that of group S1, and that there was no significant difference (*p* > 0.05) between the blood glucose content of groups S2 and S3. The results showed that low-temperature stress had a significant effect on the blood glucose index in fish, which was consistent with the results in grouper (*Epinephelus* spp.), largemouth bass (*Micropterus salmoides*), and turbot (*Psetta maxima*) [10,51,52], and further supported the idea that the cooling process induced a stress reaction in the crucian carp body. 

The blood glucose concentration of fish indicates glucose metabolism as well as the fish stress response: a higher blood glucose content indicates faster glucose metabolism and vehement stress response [18]. In the cooling process, the crucian carp has to accelerate glucose metabolism to produce more energy to maintain body temperature. As glycogen in the crucian carp is converted into blood glucose, a faster cooling process may induce the crucian carp to prepare more blood glucose, thus resulting in the blood content of the groups taking the order S1 > S2 > S3 > S4. This may be why the blood glucose content of crucian carp in groups S1, S2, and S3 was significantly higher than that in group S4 (control). For the S1 group, glycogen breakdown into the blood may be prompted by the faster cooling rate causing a vehement stress response in the fish, which led to a higher rise in blood glucose levels during the cooling process. This may be why the blood content in group S1 was significantly (*p* < 0.05) higher than in the others.

#### 3.3.2. Lactic Acid Content of Crucian Carp

To investigate the effect of the cooling process on the anaerobic metabolism of the crucian carp, the content of lactic acid was determined (Figure 5B). The results in Figure 5B indicate that the lactic acid content of the crucian carp meat increased significantly (*p* < 0.05) with an increase in the cooling speed. The lactic acid content of the crucian carp meat was, in order by group, S1 > S2 > S3 > S4, i.e., the lactic acid content of group S1 was significantly (*p* < 0.05) higher than that of groups S2, S3, and S4. The lactic acid content of group S2 was significantly (*p* < 0.05) higher than that of group S3. The lactic acid content of group S4 was significantly (*p* < 0.05) lower than that of the other groups. 

In the cooling process, low temperature may induce the crucian carp to upregulate anaerobic metabolism, with the blood glucose decomposed into lactic acid. This may be the reason why the lactic acid content of groups S1, S2, and S3 was significantly (*p* < 0.05) higher than that of group S4. The cooling speed of the groups was in the order S1 > S2 > S3, which is consistent with the lactic acid content of group S1 being significantly (*p* < 0.05) higher than that of group S2, and the lactic acid content of group S2 being significantly (*p* < 0.05) higher than that of group S3. A similar phenomenon was reported when the meagre (Argyrosomus regius) was stored in a cold environment; it used muscle and liver glycogen to conduct anaerobic metabolism, and thus showed increased lactic acid content [53]. Therefore, the lactic acid content results in the crucian carp further supported the existence of anaerobic metabolism in the crucian carp body.

#### 3.3.3. Muscle Fiber Diameter of Crucian Carp

The muscle fiber diameter is related to the tenderness of fish: when myogenic fibers are more firmly combined, a larger muscle fiber diameter suggests poor tenderness [54]. The results of the muscle fiber diameter of the crucian carp (Figure 5C) showed that there was no significant (*p* > 0.05) difference among groups S1, S2, S3, and S4. This suggests that the cooling processing had little influence on the muscle fiber diameter and tenderness of the crucian carp meat. This result is consistent with the fact that there was no significant (*p* > 0.05) difference among the hardness in groups S1, S2, S3, and S4 (Table 3). 

The main reason for the change in muscle fiber diameter is endocrine hormones [55]. The fact that there was no significant difference in the muscle fiber diameter among all groups indicated that the cooling processing had little effect on the changes in endocrine hormones of the crucian carp, but aroused a greater stress reaction and anaerobic metabolism in the crucian carp. This was supported by the fact that the content of blood glucose (Figure 5A) and lactic acid (Figure 5B) of the crucian carp treated with cooling processing was significantly higher than that of the control (group S4).

## 4. Conclusions

The results of the respiratory rate and survival time indicated that the DDT of the crucian carp was 1.6 °C. The effect of cooling speed showed a significant (*p* < 0.05) influence on the edible quality of the crucian carp; a faster cooling speed resulted in a lower pH, L* value, a* value, gumminess, springiness, cohesiveness, stickiness, chewiness, CMP, and UMP content for the crucian carp meat, and thus the faster cooling speed resulted in a lower sensory evaluation score. A possible reason for the decrease in meat quality of the crucian carp is that a faster cooling speed led to a vehement stress response and higher anaerobic metabolism in the crucian carp body. This was supported by the fact that the contents of blood glucose and lactic acid in the crucian carp treated with a higher cooling speed were significantly (*p* < 0.05) higher than those of the control. Combining all the results of the cooling speed and the eating quality of the crucian carp meat, it was found that a cooling speed of 2 °C/h followed by 1 °C/h was better than others, not only causing little degradation in the eating quality of the crucian carp meat, but also maintaining a longer survival time at the DDT for the crucian carp.

## Figures and Tables

**Figure 1 foods-12-00792-f001:**
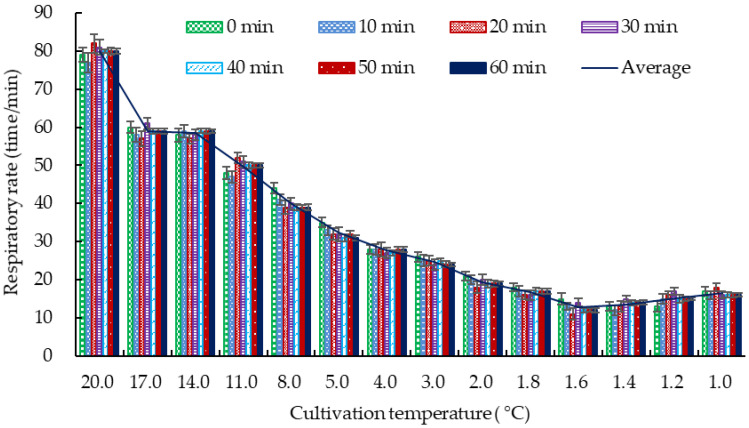
Effect of temperature on the respiratory rate of crucian carp.

**Figure 2 foods-12-00792-f002:**
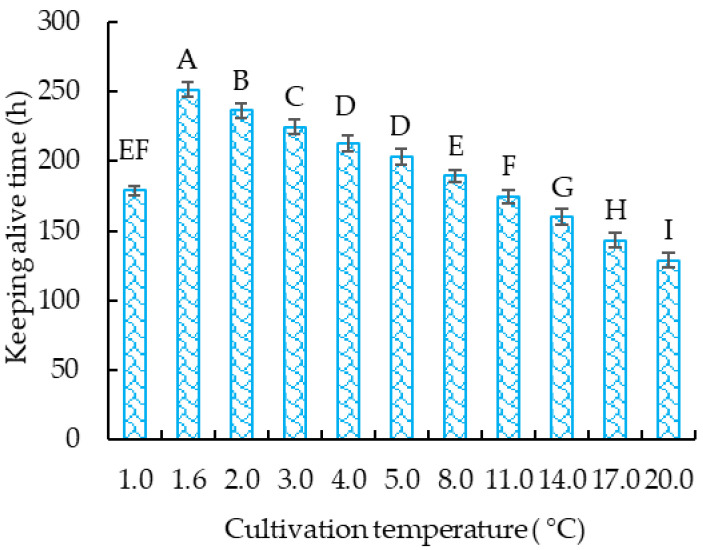
Effects of different temperatures on the survival time of the crucian carp. Different letters above the standard deviation represent significant differences between the groups (*p* < 0.05).

**Figure 3 foods-12-00792-f003:**
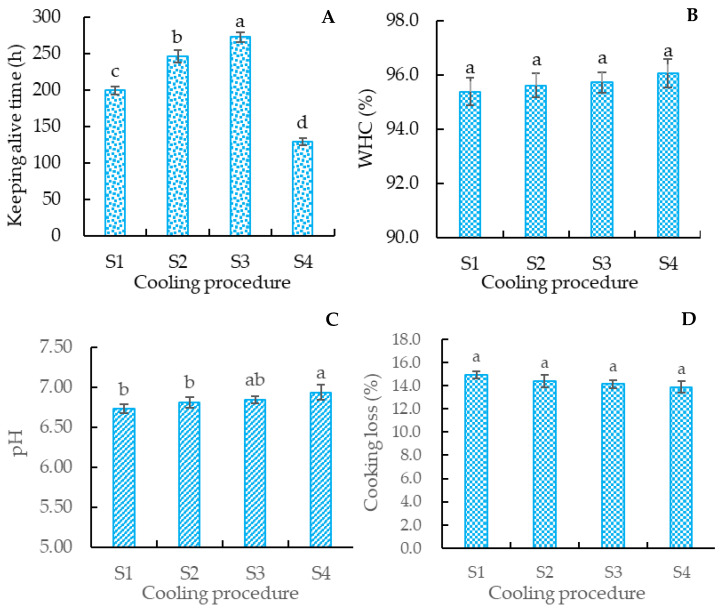
Effects of cooling speed on Keeping alive time, WHC, pH, and cooking loss of crucian carp meat. (**A**) Keeping alive time, (**B**) WHC, (**C**) pH, (**D**) cooking loss. S1: treated with cooling speed of 5 °C/h followed by 1 °C/h; S2: treated with cooling speed of 3 °C/h followed by 1 °C/h; S3: treated with cooling speed of 2 °C/h followed by 1 °C/h; S4: control. Different letters above the standard deviation represent significant differences between the groups (*p* < 0.05).

**Figure 4 foods-12-00792-f004:**
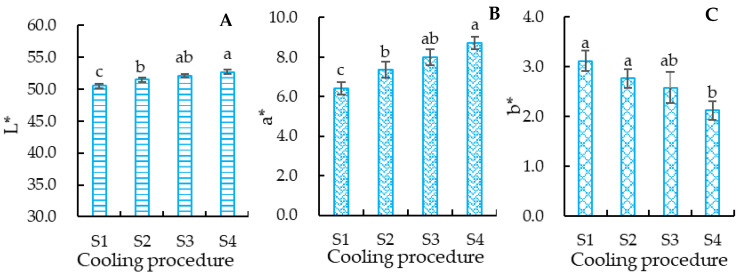
Effects of cooling speed on the color of the crucian carp meat. (**A**) Brightness value (L*); (**B**) redness value (a*); (**C**) yellowness value (b*). S1: treated with cooling speed of 5 °C/h followed by 1 °C/h; S2: treated with cooling speed of 3 °C/h followed by 1 °C/h; S3: treated with cooling speed of 2 °C/h followed by 1 °C/h; S4: control. Different letters above the standard deviation represent significant differences between the groups (*p* < 0.05).

**Figure 5 foods-12-00792-f005:**
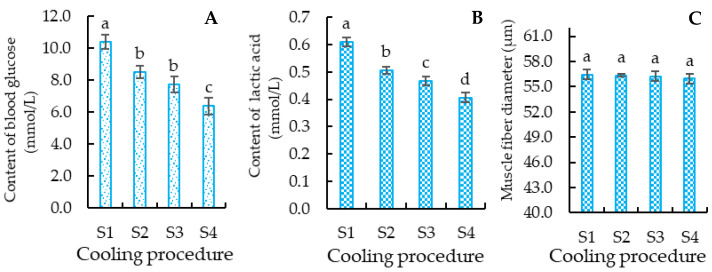
Effects of cooling speed on content of blood glucose (**A**), lactic acid (**B**), and the muscle fiber diameter (**C**) of the crucian carp. S1: treated with cooling speed of 5 °C/h followed by 1 °C/h; S2: treated with cooling speed of 3 °C/h followed by 1 °C/h; S3: treated with cooling speed of 2 °C/h followed by 1 °C/h; S4: control. Different letters beside the standard deviation represent significant differences between the groups (*p* < 0.05).

**Table 1 foods-12-00792-t001:** Definition of sensory parameters used in the quantitative descriptive analysis of the crucian carp meat.

Items	Definition	Score
Appearance	The fish has a boiled, fresh meat appearance	0–10 score (0 is the worst; 10 is the best)
Odor	Aroma intensity of boiled fish meat	
Taste	Intensity of meaty taste	
Sensory texture	Restitution ability after pressing and chewing hardness	

**Table 2 foods-12-00792-t002:** Respiratory rate and behavior of crucian carp at different temperatures.

Temperature (°C)	Respiratory Rate (Breaths/min)	Crucian Carp Behavior
20.0	82.0 ± 1.0 ^a^	Swimming around with normal breathing
17.0	59.0 ± 2.0 ^b^	Slow breathing with normal swimming
14.0	58.0 ± 1.0 ^b^	
11.0	50.0 ± 2.0 ^c^	Decreased vitality, slow swimming, unresponsiveness
8.0	39.0 ± 1.0 ^d^	
5.0	31.0 ± 2.0 ^e^	Irregular breathing, more vigorous swimming, head swaying
4.0	27.0 ± 1.0 ^f^	Wandering around, distressed
3.0	24.0 ± 1.0 ^f^	Slow breathing, sticking to the bottom of the tank, basically not swimming
2.0	19.0 ± 2.0 ^g^	Slow breathing, keeping balance, staying still but can swim with stimulation
1.0	17.0 ± 1.0 ^g^	Weak breathing, rolling over, sedate, but can swim with stimulation
0.5	6.0 ± 1.0 ^h^	Weak breathing, rolling over, no response to stimulation
0.0	3.0 ± 2.0 ^h^	Bending into an arched shape, very weak breathing, dead within a short time

Different letters beside the standard deviation represent significant differences between the groups (*p* < 0.05).

**Table 3 foods-12-00792-t003:** Effects of different cooling rates on meat texture of crucian carp.

Project	S1	S2	S3	S4
Hardness (g)	1976.87 ± 127.43 ^a^	2031.25 ± 138.02 ^a^	2065.25 ± 152.43 ^a^	2124.32 ± 154.84 ^a^
Gumminess (g·s)	−3.94 ± 0.12 ^c^	−3.79 ± 0.18 ^bc^	−3.54 ± 0.13 ^ab^	−3.24 ± 0.17 ^a^
Springiness	0.73 ± 0.01 ^b^	0.74 ± 0.01 ^ab^	0.75 ± 0.01 ^ab^	0.77 ± 0.02 ^a^
Cohesiveness	0.48 ± 0.03 ^b^	0.51 ± 0.02 ^ab^	0.52 ± 0.03 ^ab^	0.54 ± 0.02 ^a^
Stickiness (g)	862.53 ± 27.53 ^b^	883.74 ± 32.53 ^ab^	896.53 ± 34.03 ^ab^	929.68 ± 33.53 ^a^
Chewiness (g)	769.59 ± 15.99 ^c^	794.15 ± 17.29 ^bc^	813.25 ± 22.99 ^ab^	845.36 ± 19.59 ^a^
Resilience	0.28 ± 0.01 ^b^	0.29 ± 0.02 ^ab^	0.30 ± 0.02 ^ab^	0.33 ± 0.02 ^a^

S1: treated with cooling speed of 5 °C/h followed by 1 °C/h; S2: treated with cooling speed of 3 °C/h followed by 1 °C/h; S3: treated with cooling speed of 2 °C/h followed by 1 °C/h; S4: control. Data are expressed as mean ± standard deviation. Different letters beside the standard deviation represent significant differences between the groups (*p* < 0.05).

**Table 4 foods-12-00792-t004:** Effects of different cooling rates on the nucleotide content of crucian carp meat.

Project	S1	S2	S3	S4
GMP (mg/100 g)	1.41 ± 0.06 ^a^	1.46 ± 0.09 ^a^	1.34 ± 0.11 ^a^	1.26 ± 0.10 ^a^
IMP (mg/100 g)	184.38 ± 6.57 ^a^	179.95 ± 3.17 ^a^	175.86 ± 7.77 ^a^	169.77 ± 4.77 ^a^
CMP (mg/100 g)	2.89 ± 0.18 ^b^	3.06 ± 0.11 ^ab^	3.13 ± 0.17 ^ab^	3.46 ± 0.27 ^a^
UMP (mg/100 g)	1.95 ± 0.08 ^b^	2.09 ± 0.13 ^ab^	2.18 ± 0.07 ^ab^	2.31 ± 0.10 ^a^
AMP (mg/100 g)	13.34 ± 0.28 ^a^	12.91 ± 0.47 ^ab^	12.75 ± 0.19 ^ab^	12.05 ± 0.38 ^b^

S1: treated with cooling speed of 5 °C/h followed by 1 °C/h; S2: treated with cooling speed of 3 °C/h followed by 1 °C/h; S3: treated with cooling speed of 2 °C/h followed by 1 °C/h; S4: control. Different letters beside the standard deviation represent significant differences between the groups (*p* < 0.05).

**Table 5 foods-12-00792-t005:** Effects of different cooling rates on the sensory evaluation of crucian carp meat.

Item	Group	Score
Appearance	S1	8.5 ± 0.1 ^b^
	S2	9.1 ± 0.1 ^a^
	S3	9.1 ± 0.1 ^a^
	S4	9.2 ± 0.1 ^a^
Odor	S1	8.4 ± 0.2 ^c^
	S2	8.6 ± 0.2 ^bc^
	S3	8.8 ± 0.2 ^ab^
	S4	9.1 ± 0.1 ^a^
Taste	S1	8.7 ± 0.2 ^b^
	S2	9.1 ± 0.2 ^b^
	S3	9.2 ± 0.2 ^a^
	S4	9.2 ± 0.1 ^a^
Sensory texture	S1	8.8 ± 0.2 ^b^
	S2	8.8 ± 0.1 ^b^
	S3	9.2 ± 0.1 ^a^
	S4	9.3 ± 0.1 ^a^

S1: treated with cooling speed of 5 °C/h followed by 1 °C/h; S2: treated with cooling speed of 3 °C/h followed by 1 °C/h; S3: treated with cooling speed of 2 °C/h followed by 1 °C/h; S4: control. Different letters beside the standard deviation represent significant differences between the groups (*p* < 0.05).

## Data Availability

Data Availability Statement: Data is contained within the article.

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
