# Peer review of "Effect of Deep Dormancy Temperature Cultivation on Meat Quality of Crucian Carp (Carassius auratus)"

_foods, 2023, doi:10.3390/foods12040792_

Round 1
Reviewer 1 Report
Due to the ethical concerns, authors should provided the approval of the Ethics Committee on the experiment conducted. Moreover, they also should clearly describe how the fish were killed.
Author Response
Question 1: I don't feel qualified to judge about the English language and style
Response:To make the English language and style consistent to Foods, this manuscript has undergone English language editing by MDPI (Invoice ID: english-54443)
Question 2: Due to the ethical concerns, authors should provided the approval of the Ethics Committee on the experiment conducted. Moreover, they also should clearly describe how the fish were killed.
Response:A document sealed by Ethic Committee of Chengdu University has been submitted along the revised manuscript. We confirm that our research complies with the commonly-accepted '3Rs', and the details of fish killing procedures have been added in lines 85-88, which was complies with the Animals (Scientific Procedures) Act 1986. Code of Practice for the Housing and Care of Animals Bred, Supplied or Used for Scientific Purposes. https://assets.publishing.service.gov.uk/government/uploads/system/uploads/attachment_data/file/388535/CoPanimalsWeb.pdf

Reviewer 2 Report
This study was performed to determine the effect of various condition of deep dormancy cultivation on biochemical properties and meat quality attributes of crucian carp. The research objective and experimental approaches were well-fitted. However, some issues mentioned below should be improved.
L43 Delete of before after.
L99 It would be better to describe “every 60 min”, if it is correct.
L121 In the experiment, it is not clear how many fishes were used as replication and repetition. Should be clarified.
L175-176 Please describe the final core temperature of muscle after cooking.
L358-359/L414-415 As mentioned, the denaturation of muscle protein is greatly associated with water-holding capacity and textural properties. However, it seems that the results for WHC and texture and related discussion were quite conflicting. Should be rephrased.
L474-483 As shown in Table 4, there was significant difference in umami compounds between S1, S2, and S3. However, the description and discussion have only focused on difference between control (S4) and other treatments.
In figure 5, it would be better to add the result of blood pH.
Author Response
Question 1: L43 Delete of before after.
Response:Done.
Question 2: L99 It would be better to describe “every 60 min”, if it is correct.
Response:The word “every” has been added before 60 min in L99.
Question 3: L121 In the experiment, it is not clear how many fishes were used as replication and repetition. Should be clarified.
Response:Done. The description in L121 has been revised as follow:
For each experiment, 12 crucian carps (3 repetition × 4 crucian carps) were used to perform triplicate experiments, and three batches (12 crucian carps per batches) were evaluated for each experiment.
Question 4: L175-176 Please describe the final core temperature of muscle after cooking.
Response:The cored temperatures of fish muscle after cooking and the cooling procedure have been added in L182- 183.
Question 5: L358-359/L414-415 As mentioned, the denaturation of muscle protein is greatly associated with water-holding capacity and textural properties. However, it seems that the results for WHC and texture and related discussion were quite conflicting. Should be rephrased.
Response:Sentences in L358-359/L414-415 were rephrased. Thank you for reminding us.
Question 6: L474-483 As shown in Table 4, there was significant difference in umami compounds between S1, S2, and S3. However, the description and discussion have only focused on difference between control (S4) and other treatments.
Response:The L474-483 have been rewritten, the description and discussion about group S1, S2, S3 and S4 have been added in lines 485-490.
Question 7: In figure 5, it would be better to add the result of blood pH.
Response:The blood pH of the fish sample was not determined. The main reasons were that (1) there was less than 1.5 ml blood in the tail of a crucian carp, which was not enough for determining the blood pH. (2) According to the Research and Publication Ethics of Foods about using of Animals in experiments, authors ensured that their research complies with the commonly-accepted '3Rs' (Replacement of animals by alternatives wherever possible; Reduction in number of animals used; AND Refinement of experimental conditions and procedures to minimize the harm to animals.
Round 2
Reviewer 1 Report
In my opinion, despite obtaining interesting results, the authors did not address ethical concerns. In the Materials and Methods section they did not state in which year and in what quantity the research material was purchased. The unnumbered statement from the Ethics Committee of Chengdu University, presented as an appendix, is signed with a date of 2020 and the full title of the submitted article. The research was conducted as part of a project that should have been approved by the Ethics Committee. The authors also do not state precisely at what point the fish were killed and when blood was drawn for testing. Paragraphs 132-133 indicate that the fish were kept in thermostats until they died.
Author Response
Question 1: In my opinion, despite obtaining interesting results, the authors did not address ethical concerns. In the Materials and Methods section they did not state in which year and in what quantity the research material was purchased.
Answer: To clearly show the purchasing time and quantity of the crucian carps for each experiment, the related details have been added in lines 98-100, 109-111, 119-122, 139-143.
Question 2: The unnumbered statement from the Ethics Committee of Chengdu University, presented as an appendix, is signed with a date of 2020 and the full title of the submitted article. The research was conducted as part of a project that should have been approved by the Ethics Committee. The authors also do not state precisely at what point the fish were killed and when blood was drawn for testing.
Answer: To address the details of killing fish and drawing blood, the method of determining blood glucose (2.5) and blood lactic acid (2.6) were rewritten in lines 149 -152, 162-165.
Question 3: Paragraphs 132-133 indicate that the fish were kept in thermostats until they died.
Answer: Paragraphs 132-133 have been rewritten, details about the fish treatment were added (lines 123 -126).